# Current Advances in the Regeneration of Degenerated Articular Cartilage: A Literature Review on Tissue Engineering and Its Recent Clinical Translation

**DOI:** 10.3390/ma15010031

**Published:** 2021-12-21

**Authors:** Farah Daou, Andrea Cochis, Massimiliano Leigheb, Lia Rimondini

**Affiliations:** 1Department of Health Sciences, Center for Translational Research on Autoimmune and Allergic Diseases-CAAD, Università del Piemonte Orientale UPO, 28100 Novara, Italy; farah.daou@uniupo.it (F.D.); andrea.cochis@med.uniupo.it (A.C.); massimiliano.leigheb@uniupo.it (M.L.); 2Department of Orthopaedics and Traumatology, “Maggiore della Carità” Hospital, 28100 Novara, Italy

**Keywords:** cartilage regeneration, tissue engineering, biomaterials, clinical translation

## Abstract

Functional ability is the basis of healthy aging. Articular cartilage degeneration is amongst the most prevalent degenerative conditions that cause adverse impacts on the quality of life; moreover, it represents a key predisposing factor to osteoarthritis (OA). Both the poor capacity of articular cartilage for self-repair and the unsatisfactory outcomes of available clinical interventions make innovative tissue engineering a promising therapeutic strategy for articular cartilage repair. Significant progress was made in this field; however, a marked heterogeneity in the applied biomaterials, biofabrication, and assessments is nowadays evident by the huge number of research studies published to date. Accordingly, this literature review assimilates the most recent advances in cell-based and cell-free tissue engineering of articular cartilage and also focuses on the assessments performed via various in vitro studies, ex vivo models, preclinical in vivo animal models, and clinical studies in order to provide a broad overview of the latest findings and clinical translation in the context of degenerated articular cartilage and OA.

## 1. Introduction

Cellular senescence is predominantly correlated with tissue aging and loss of function, but its physiological role must be indeed evaluated in relation to the stage of life. In early life, cellular senescence contributes to the attenuation of tissue damage, promotion of wound healing, and suppression of tumorigenesis; on the contrary, in old age, it fosters inflammation, aging, and aging-related diseases and also limits tissue regenerative potential [1]. In addition to the latter, the accumulation of senescent cells is systemic and continuous, and this results in a persistent imbalance in the homeostasis of almost all tissues. One of the organ systems that is significantly affected by cellular senescence is the musculoskeletal system, since it results in bone, muscle, and cartilage degeneration [2].

Nowadays, cartilage degeneration continues to be a major challenge for clinicians due to the limited self-healing properties originating from the low proliferation ability of chondrocytes, the sole cellular constituents of cartilage [3], as well as from the absence of blood vessels, nerves, and lymphatics [4]. The most commonly documented cartilage degeneration is that of articular cartilage covering synovial joints [3]. In this regard, several studies showed that articular cartilage degeneration in the knees is prevalent in both symptomatic and asymptomatic people. In one study, chondral and osteochondral defects were observed in 61% of arthroscopies of patients with symptomatic knees [5]; additionally, another study involving athletes indicated that more than 50% of asymptomatic athletes displayed full-thickness focal chondral defects [6]. This high prevalence of articular cartilage degeneration represents a major clinical challenge, since, in the long-term, articular cartilage degeneration can contribute to the development of osteoarthritis (OA) (a multifactorial joint condition) [7], which is the third most common musculoskeletal disorder requiring rehabilitation after low back pain and fractures [8]. Moreover, the relatively high prevalence of articular cartilage degeneration and the predisposition to OA are further aggravated by the aging global population with an estimated 727 million persons aged 65 years or over in 2020 and a projected increase to reach over 1.5 billion by 2050 [9]. As a result, age-related changes in articular cartilage and OA will soon represent a major public health challenge, as well as a major economic burden in many countries.

Therapeutic approaches for treating articular cartilage degeneration are diverse and can be categorized into three types: (i) symptomatic, (ii) reparative (or restorative), and (iii) regenerative. Symptomatic treatments include pain killers, anti-inflammatory drugs, and intra-articular injections of corticosteroids, hyaluronic acid (HA), or platelet-rich plasma (PRP); reparative treatments are microfracture, abrasion, drilling, osteochondral allograft, and mosaicplasty; whereas regenerative treatments entail autologous chondrocyte implantation (ACI) and matrix-induced autologous chondrocyte implantation (MACI) [10,11,12]. The reparative and regenerative treatments are reported to be effective in treating chondral and subchondral defects, but they can neither cure nor slow down articular cartilage degeneration; hence, tissue engineering strategies were introduced to overcome these limitations. Since its inception, articular cartilage tissue engineering yielded positive results and experienced major advancements, and consequently, it is now considered a promising alternative for replicating the structure and function of native articular cartilage [10,13].

This literature review first describes the changes in articular cartilage at the cellular and extracellular matrix (ECM) levels relative to the stages of articular cartilage degeneration, and then highlights the latest advances in tissue engineering strategies applied for articular cartilage regeneration, including cell-based and cell-free ones. It also includes the assessments performed via various in vitro studies, ex vivo models, preclinical in vivo animal models, and clinical studies. In particular, this review is aimed to provide a critical update of the current status of articular cartilage tissue engineering and its clinical translation relating to cartilage degeneration and OA.

## 2. Materials and Methods

A literature search of studies on articular cartilage tissue engineering was conducted in the PubMed database with the following Medical Subject Heading terms: (Cartilage, Articular[MeSH Terms]), (Regeneration[MeSH Terms]), (Tissue Engineering[MeSH Terms]), and (Biocompatible Materials[MeSH Terms]) to encompass the relevant literature. All English-language experimental, observational, and interventional studies published between January 2011 and October 2021 were included for evaluation, whereas conference abstracts and review articles were excluded. One author (F.D.) screened article titles and abstracts in the initial search to identify relevant studies. Subsequently, the full text of every article was read by each reviewer (A.C. and L.R. read preclinical studies, and M.L. read clinical studies) to shortlist articles. For preclinical studies, the most recent articles were cited when similar research was performed by other research groups. For clinical studies, the work targeting human subjects with articular cartilage lesions, osteochondral lesions, and OA was included, and anatomical/imaging and clinical outcomes were reported. In total, 49 experimental studies and 15 clinical studies were included in this literature review. Considering the heterogeneity of the selected papers, the results were reported only in a descriptive manner without any qualitative-quantitative comparison.

## 3. Age-Related Changes in Articular Cartilage

Articular cartilage is classified as hyaline cartilage, and it is mainly composed of chondrocytes and a dense ECM produced by them. Its thickness ranges between 2 to 4 mm; however, it exhibits structural and compositional heterogeneity across four different zones, which are the (i) superficial (tangential), (ii) middle (transitional), (iii) deep, and (iv) calcified zones [14]. The structure of healthy articular cartilage is schematized in Figure 1 [15].

Age-related changes in articular cartilage involve both components (chondrocytes and ECM), and the progressive nature of pathological conditions makes their classification essential, not only to detect the extent of the defects but also to guide clinical decision-making. The most commonly applied classification method is the arthroscopic grading system developed by the International Cartilage Repair Society (ICRS), which divides defects into four grades [16], as detailed below and schematized in Figure 2:ICRS Grade 0: Normal;ICRS Grade 1: Nearly Normal (superficial lesions);ICRS Grade 2: Abnormal (lesions extending down to <50% of cartilage depth);ICRS Grade 3: Severely Abnormal (cartilage defects extending down >50% of cartilage depth);ICRS Grade 4: Severely Abnormal (cartilage defects extending through the subchondral bone).

### 3.1. Age-Related Changes in Chondrocytes

Chondrocytes are specialized cells constituting only 2% of the total volume of articular cartilage and exhibiting variations in shape, number, and size across its zones. They play a central role in the synthesis and maintenance of a normal ECM, thus, having an adequate number of functionally competent chondrocytes is of crucial importance for articular cartilage homeostasis [14].

The age-related changes in chondrocytes involve a decrease in the cell density (mainly in the superficial zone) and cellular dysfunction [3]. These changes are mediated by two mechanisms: first, chondrocyte senescence (intrinsic and/or extrinsic), which is associated with different types of cell death and impaired responses to extracellular stimuli [17]; second, the downregulation of the anabolic cell signaling due to the decrease in the levels of growth factors, such as transforming growth factor-β (TGF-β2 and TGF-β3) and bone morphogenetic protein-7 (BMP-7) [3,17]. Moreover, senescent chondrocytes exhibit a senescence-associated secretory phenotype (SASP) [3] characterized by increased production of pro-inflammatory cytokines, matrix metalloproteinases (MMPs), and growth factors [17]. Another age-related change in chondrocytes is mediated by advanced glycation end-products (AGEs), which are produced by the spontaneous nonenzymatic glycation of proteins during physiological aging [3,17]. Studies showed that AGEs can interact with their specific cell surface receptors, known as receptors for advanced glycation end-products (RAGE), which are expressed by chondrocytes and upregulated during aging [18]. This interaction activates chondrocyte RAGE and has been shown to stimulate chondrocyte hypertrophy [19] and catabolic signaling pathways [20].

### 3.2. Age-Related Changes in Cartilage ECM

The ECM is responsible for the mechanical properties of articular cartilage. Its functions are mediated by two phases: the liquid phase that is composed mainly of water and inorganic ions and accounts for 65 to 80% of the wet weight, and the solid phase that is primarily made up of collagens (mainly type II collagen) and proteoglycans (including aggrecan, decorin, biglycan, and fibromodulin) and constitutes the remaining dry weight [14].

The hallmarks of articular cartilage degradation are the changes in the total amount and composition of the ECM [3]. These changes are mediated by three mechanisms: first, chondrocyte death and dysfunction (discussed above); second, accumulation of apoptotic bodies; third, marked increase in AGEs [3,17]. Chondrocyte apoptosis in the absence of phagocytic cells leads to the accumulation of apoptotic bodies that leads to pathologic cartilage calcification [3], which is suggested to prompt increased fibrillation and OA [21]. AGEs, on the other hand, cause an increase in the cross-linking of collagen molecules that adversely impacts the biomechanical properties of cartilage [22].

## 4. Tissue Engineering Strategies for Articular Cartilage Regeneration

Extensive attempts have been made to engineer cartilage tissues with structural and functional properties similar to those of native articular cartilage. These efforts have brought major advancements in the field of articular cartilage tissue engineering and, at the same time, generated a heterogeneity in the biomaterials, biofabrication, and assessments applied in this field of research [23,24]. In the following sections, the most recent advances in cell-based and cell-free tissue engineering of articular cartilage in the context of aging and OA are presented, along with their assessments via in vitro, ex vivo, and preclinical in vivo studies, as well as their current clinical translation (clinical studies).

### 4.1. Cell-Based Tissue Engineering Strategies

Research on cell-based articular cartilage tissue engineering has identified new cell sources, scaffolds, and bioactive molecules, as well as novel composites combining these components. In this section, cell-based tissue engineering is divided into scaffold-based strategies, scaffold-free strategies, and injectable materials.

#### 4.1.1. Scaffold-Based Strategies

The use of scaffolds in articular cartilage tissue engineering represents a largely applied strategy, since scaffolds are intended to support cellular proliferation and differentiation, as well as to deliver pro-chondrogenic bioactive molecules, and in that pursuit, multiple scaffolds were designed for various purposes. In one of the most recent studies, kartogenin (KGN), which is a small molecule discovered by Johnson et al. (2012) to be a chondrogenic and chondroprotective agent [25], was utilized by Teng et al. (2021). This group optimized the widely used combination of HA polymer and human bone marrow-derived mesenchymal stem cells (BM-MSCs) by synthesizing a methacrylated hyaluronic acid (MeHA), integrating two biomimetic peptides (arginine-glycine-aspartate (RGD) and histidine-alanine-valine (HAV)) to promote cellular activity, and introducing KGN-encapsulated poly(lactic-co-glycolic acid) (PLGA) microspheres (KGN@PM) to deliver KGN and induce the differentiation of BM-MSCs to chondrocytes. The functionalized HA hydrogels were evaluated via studies in vitro and in vivo (subcutaneous implantation into male nude mice). The results showed that this hydrogel system enhanced proliferation, adhesion, condensation, and chondrogenic differentiation of human mesenchymal stem cells (hMSCs), mainly when both peptides (RGD and HAV) were added [26]. Shen et al. (2020) also worked on biomaterials extensively used in orthopedic tissue engineering applications, which are the polylactic acid (PLA)-based ones, to improve the delivery of the TGF-β. First, a photopolymerizable poly-D, L-lactic acid/polyethylene glycol (PDLLA) hydrogel was mechanically reinforced by incorporating graphene oxide (GO) nanosheets, and then the GO/PDLLA scaffold was loaded with TGF-β3 and human BM-MSCs. Both studies in vitro and in vivo (subcutaneous implantation into non-obese diabetic/severe combined immunodeficient (NOD/SCID) female mice) exhibited sustained release of TGF-β3 and verified the role of GO nanosheets on improving the mechanical strength of the scaffold and chondrogenesis [27].

On the other hand, researchers are studying the application of innovative strategies, such as laser technology, to improve the scaffolds used in cartilage tissue engineering. Nürnberger et al. (2021) focused on developing a scaffold with a defined architecture that properly guides collagen alignment during articular cartilage regeneration. To achieve their goal, they first engraved human articular cartilage ECM using a CO_2_ laser in three patterns (hole, line, and grid), then decellularized (with subsequent glycosaminoglycan (GAG) depletion) or devitalized the matrix, and finally seeded it with different types of cells for analysis in vitro and in vivo (subcutaneous implantation of bovine osteochondral cylinders into female athymic NMRI nude mice). The matrix with grid patterns was selected for its preferable properties, and the decellularized GAG-depleted one exhibited superior ability to guide the newly synthesized matrix towards a vertical alignment, avoiding an undesired random deposition, and thus, improving the neo-cartilage regeneration [28].

While some researchers have been optimizing routinely used scaffolds, others have been exploring the application of new components in articular cartilage tissue engineering. Kim et al. (2021) in their recent research tested recombinant human transglutaminase 4 (rhTG-4) in vitro and in vivo using New Zealand white rabbits with full-thickness osteochondral defects for the treatment of osteochondral defects by supplementing a system composed of synovium-derived mesenchymal stem cells (SMSCs) encapsulated in HA/collagen/fibrinogen composite scaffold. The results supported the role of the rhTG-4 in enhancing articular cartilage regeneration by upregulating integrin β1 and actin remodeling [29]. A new study suggested the human dermal-derived collagen as a scaffold with potential application in articular cartilage tissue engineering. In this study, Dang et al. (2021) fabricated a complex of human adipose-derived mesenchymal stem cells (AD-MSCs), dermal-derived collagen matrix sheet, and collagen substrates and observed marked effects of dermal collagen on the in vitro expansion of functional chondrocytes [30]. A similar strategy was applied by Özdemir et al. (2020) using decellularized human placenta as a scaffold. In their work, this novel matrix was incorporated with human BM-MSCs and/or PRP, and then assessed in vivo using Wistar albino rat models with osteochondral defects. However, results showed that the efficacy for articular cartilage regeneration is comparable for the scaffold alone and the scaffold augmented with BM-MSCs, PRP, or both [31].

Similar efforts were made by Barlian et al. (2020), who fabricated a silk spidroin-fibroin scaffold, and then loaded it with human Wharton’s Jelly mesenchymal stem cells (hWJ-MSCs) and a chondrogenesis inducer (L-ascorbic acid (LAA) or PRP). By means of in vitro studies, they detected the adequate weight to weight (*w*/*w*) concentration of silk spidroin (SS) and silk fibroin (SF), as well as the most effective and safe concentrations of LAA and PRP, and the results showed that the scaffold composed of 10% (*w*/*w*) SS and 90% (*w*/*w*) SF has superior structural and mechanical properties, and supplementation with 10% PRP is optimal for inducing chondrogenic differentiation of hWJ-MSCs [32]. In a recent work, tanshinone IIA (TAN), which is an active ingredient extracted from Danshen root (*Salvia miltiorrhiza* Bunge) and possessing anti-inflammatory [33], antioxidative [34], and anti-apoptotic [35] properties, was used with SF scaffolds. In their work, Chen et al. (2020) coupled SF scaffolds with TAN and seeded them with New Zealand white rabbit chondrocytes. Assessments of the novel scaffold in vitro and in vivo using nude mice (subcutaneous implantation) and New Zealand white rabbit models with full-thickness cartilage defects revealed that chondrocytes in these scaffolds can produce hyaline-like cartilage and thus promote cartilage regeneration better than SF scaffolds alone, mainly when the concentration of TAN is 10 μg/mL [36].

Other researchers harnessed self-assembling peptides, which are composed of amino acid sequences that assemble to form nanostructures (nanofibers, hydrogels, etc.) for reinforcing scaffolds. For instance, a new platform was designed and tested in vitro by Rubí-Sans et al. (2019) in an attempt to detect the role of RAD16-I in reinforcing a system of poly(ε-caprolactone) (PCL) scaffold and human BM-MSCs for articular cartilage regeneration, and the outcomes were promising regarding mechanical properties and chondrogenic differentiation [37]. Another self-assembling peptide, IEIK13, was used by Dufour et al. (2021) along with fibrin and articular chondrocytes (from human and cynomolgus macaques) treated with a pro-chondrogenic cocktail. This new system was tested in vitro and in vivo using for the very first time a non-human primate model (cynomolgus macaques) with full-thickness articular cartilage defects, and the results showed that the chondrocyte-loaded IEIK13 and acellular IEIK13 scaffolds are equally effective in repairing cartilage defects [38].

Another recent improvement of cell-based, scaffold-based articular cartilage tissue engineering approaches is the deployment of co-culturing, since cell signaling can have a crucial impact on counteracting the poor regenerative properties of articular cartilage by maintaining chondrocyte phenotype and promoting cartilage ECM regeneration. The first co-culture system to be discussed is composed of MSCs and chondrocytes. Scalzone et al. (2019) used an optimized, thermo-sensitive chitosan, and β-glycerophosphate (BGP) hydrogel as an in vitro 3D scaffold to encapsulate BM-MSCs, and then loaded human articular cartilage chondrocytes (ACCs) spheroids on the top of the composite. In addition to the favorable cytocompatibility evaluation, the co-culture system exhibited promising neocartilage regeneration, suggesting a potential role of MSCs in positively influencing the metabolic activity of chondrocytes that is typically reduced in injured sites [39]. The second type of co-culture systems employs chondrons (composed of chondrocytes and the surrounding pericellular matrix (PCM)) coupled with other cells. One of the latest research works on chondrons was conducted by Duan et al. (2021). In their work, they used rabbit models to isolate chondrocytes and chondrons, cultured cells alone or together, and finally, encapsulated cells in alginate spheres. The efficacy of the co-culture system in comparison with chondrocytes and chondrons alone was evaluated in vivo by filling knee osteochondral defects of New Zealand white rabbit models with one of the three different tissue engineered constructs. Although the co-culture system exhibited satisfactory collagen type II (Col-2), aggrecan (AGG), and GAG production, as well as cartilage repair, the obtained results were comparable to those acquired with chondrocytes alone [40]. Contrarily, Owida et al. (2017) tested a co-culture system of MSCs and chondrons in vitro, with three types of cultures established as follows: bovine chondrocytes, bovine chondrons, and bovine chondrons and rat MSCs. Evaluations of cartilage ECM production showed that the xenogeneic co-culture system was superior in comparison to the single-cell cultures [41].

Despite the promising results of the studies discussed above, they all omit the replication of the topographical zonal organization of articular cartilage; therefore, several groups of researchers have been working on the development of scaffolds with a biomimetic gradient similar to native articular cartilage. A recent work by Castilho et al. (2019) resulted in the fabrication of a hydrogel with zonal mechanical properties similar to those of the articular cartilage by reinforcing gelatin–methacrylamide (GelMA) hydrogels with a bi-layered microfiber architecture. When equine chondrocytes were embedded in the novel scaffolds in vitro, neo-cartilage formation was observed in static conditions after TGF-ß1 supplementation and under mechanical stimulation without the TGF-ß1 supplementation [42]. Additionally, Zhu et al. (2018) fabricated a tissue-scale stiffness gradient hydrogel as a 3D cell niche to guide the regeneration of articular cartilage with topographical organization. In their work, a photocrosslinkable, multi-arm polyethylene glycol (PEG) hydrogel system composed of PEG-norbornene, PEG-dithiol, and methacrylated chondroitin sulfate (CS-MA) was loaded with neonatal bovine chondrocytes or hMSCs and then evaluated in vitro. The outcomes revealed that such hydrogels are capable of influencing cell behavior, as well as new cartilage ECM deposition in a zone-dependent manner [43]. A similar work was previously proposed by Steele et al. (2014), who used electrostatic deposition to laminate a bilayered PCL scaffold (composed of a fiber zone and a porous zone) to a bulk porous particulate-templated scaffold produced with 0.03 or 1.0 mm^3^ porogens. When the scaffolds were loaded with bovine chondrocytes in vitro, smaller porogens were superior in supporting chondrogenesis [44]. Although the results obtained by these studies are surely encouraging, their limitation is the absence of evaluations via in vivo studies.

#### 4.1.2. Scaffold-Free Strategies

Scaffold-free tissue engineering for articular cartilage regeneration represents a promising alternative for overcoming the limitations associated with scaffold-based tissue engineering, mainly regarding the long-term safety of the devices themselves. The most commonly used technology for this purpose is the so-called cell sheet technology consisting of implantable artificial proto-tissues composed of cells in high-density and tightly interconnected to each other by a dense ECM that is harvested avoiding the use of enzymes by thermo-responsive substrates [45,46]. One of the latest research works on cell sheet technology was conducted by Wongin et al. (2021) in an attempt to evaluate an osteochondral-like tissue composed of human chondrocyte sheets cultivated onto human freeze-dried cancellous bone. For this purpose, in vitro studies and in vivo studies using New Zealand white rabbit models with knee osteochondral defects were performed, and the latter showed that chondrocyte sheets support the formation of hyaline-like cartilage and chondrocyte sheet-cancellous bone tissues improve the osteochondral repair [47]. Alternatively, cell sheets of human chondrocytes and synoviocytes were tested by Takizawa et al. (2020), who transplanted cell sheets of human chondrocytes, human synoviocytes, or both into immunodeficient rats with knee osteochondral defects. The results of these in vivo studies showed that the number of cells decreased in all groups after 12 weeks, and only chondrocyte sheets were able to fill the defects with a combination of hyaline cartilage and fibrocartilage, and not only with fibrous tissue [48].

However, the cell sheet technology has also been extensively applied with MSCs, and not only with differentiated cells. Thorp et al. (2020) developed scaffold-free, pre-differentiated, hyaline-like cell sheet constructs in vitro from BM-MSCs (grown in chondrogenic medium for 3 weeks) using the cell sheet technology. The constructs were examined via in vitro studies and ex vivo studies using human articular cartilage pieces, and these confirmed chondrogenic differentiation, maintenance of hyaline-like chondrogenic phenotypes, and spontaneous adhesion to the cartilage surfaces [49]. Another attempt of the cell sheet technology was performed by You et al. (2020) using human amniotic mesenchymal stem cells (hAMSCs). In this research, hAMSCs were used to fabricate cell sheets, which were then enriched by the addition of cartilage particles, and finally, evaluated in vivo using New Zealand white rabbit models with knee osteochondral defects. The outcomes revealed that hAMSC sheet–cartilage particle complexes exhibit promising macroscopic, histological, as well as cartilage and subchondral bone regeneration properties [50].

#### 4.1.3. Injectables

In view of a possible clinical application, researchers are aspiring to utilize technologies that enable shifting from invasive surgical procedures to minimally invasive or non-invasive ones for the regeneration of articular cartilage. Out of the numerous developed injectables, the simplest ones are aimed at delivering cells only into the defect site. For instance, cell sheet technology was applied by Wasai et al. (2021) to fabricate injectable allogeneic polydactyly-derived chondrocyte sheets (PD) cell sheet fragments rather than large cell sheets that require invasive surgery. The results showed that there are no significant differences between the PD sheets and the PD sheets-mini in terms of cell count and viability, the number of humoral factors produced, and the histological characteristics; in addition, the injection of the PD sheets-mini did not alter cell viability [51]. Moreover, in a one-of-a-kind study, Takagi et al. (2020) investigated whether weekly intra-articular injections of autologous AD-MSCs cell sheets (using a culture medium supplemented with ascorbate-2-phosphate) can attenuate the progression of OA in vivo using a rabbit anterior cruciate ligament transection (ACLT) model. Not surprisingly, the delivered AD-MSCs exhibited protective properties towards chondrocytes, thus preventing cartilage degeneration, and also resulted in milder progression of OA in comparison with the control group [52]. An alternative work by Köhnke et al. (2021) involved the evaluation of AD-MSCs injection for the treatment of temporomandibular joint osteoarthritis (TMJ-OA) in vivo using New Zealand white rabbit models. Based on these premises, animal models were randomized to receive one of the following four different injections: (i) AB serum, (ii) HA, (iii) stem cells, or (iv) stem cells loaded in HA. The best outcomes regarding articular cartilage regeneration after a 4-week follow-up were observed with stem cells, especially those embedded in HA; however, there were no significant differences among the four groups in terms of tissue porosity and heterogeneity of mineralization [53]. A similar injectable was also tested by Qu et al. (2021), and the aim was to deliver BM-MSCs using open-porous PLGA microspheres via alkaline treatment, and the results of both in vitro studies and in vivo studies using Sprague–Dawley rat osteochondral defect models confirmed improved articular cartilage regeneration [54]. In a previous study by Prasadam et al. (2018), authors aimed to deliver a co-culture of MSCs and chondrocytes as an injectable. For evaluating this co-culture system, in vitro studies, ex vivo studies using a cartilage defect model implanted subcutaneously in NOD-SCID mice, as well as in vivo studies using Wistar rat models with knee OA were performed. The obtained results showed that the co-culture system was more effective in promoting cartilage regeneration, as well as decreasing fibrosis [55].

However, multi-factor injectables were also developed by some researchers. In a new study by Co et al. (2021), injectable biomaterials were implemented in preventive medicine, and the goal was to counteract post-traumatic osteoarthritis (PTOA) via click chemistry. For this purpose, they developed a dual-acting system that first targets apoptotic chondrocytes by a PEG polymer carrier conjugated with apoptosis-targeting peptide-1 (ApoPep-1) and trans-cyclooctene (TCO) and, second, delivers metabolically active chondrocytes. The results of the in vitro assessments and ex vivo studies using human cartilage explant PTOA models showed promising outcomes in specifically targeting and treating cartilage injury [56]. Moreover, more complex technologies were applied to optimize articular cartilage regeneration via bioactive molecules. Very recent research by Xu et al. (2021) investigated a unique tissue engineering strategy for the treatment of OA, where engineered exosomes were synthesized to encapsulate KGN and to express the MSC-binding peptide E7. These exosomes were mixed with synovial fluid-derived mesenchymal stem cells (SF-MSCs) derived from an OA patient and evaluated by means of in vitro studies and in vivo studies using Sprague–Dawley rat models with knee OA, which validated the superiority of the targeted KGN delivery system in the cartilage regeneration process [57].

#### 4.1.4. Clinical Studies

According to the applied search criteria, the studies reported above represent the most recent and representative research work on cell-based tissue engineering for articular cartilage regeneration. Despite accounting for major milestones in the field of articular cartilage tissue engineering, translating the findings of these studies to clinical practice is impeded by the limited number of clinical studies. Below are the findings of the clinical studies on cell-based tissue engineering strategies for articular cartilage regeneration.

The most recent results of a randomized clinical trial (RCT) are those on the safety and efficacy of the infrapatellar fat pad (IPFP), which is routinely removed in knee arthroscopic surgery, as a source of MSCs reported by Zhou et al. (2021). Patients with symptomatic articular cartilage lesions in the knee (Kellgren and Lawrence (KL) Grade ≤ level 3) were randomly assigned into two groups: (i) knee arthroscopic therapy only or (ii) knee arthroscopic therapy with autologous IPFP cell concentrates. The outcomes supported the positive effect of IPFP cell concentrates on reducing pain and improving articular functionality in these patients [58]. Another RCT by Qiao et al. (2020) evaluated the efficacy of microfracture alone or in combination with the injection of HA or HA loaded with autologous human adipose-derived mesenchymal progenitor cells (haMPCs). These three different treatment strategies were randomized over 30 patients with medial femoral-tibial condylar or trochlear-patellar cartilage defects associated with moderate to severe (KL Grade 3) knee OA. Preliminary results showed that microfracture along with HA and autologous haMPCs injection can result in long-term clinical improvement (more than 12 months post-surgery), but the regenerated tissues were characterized by the presence of fibrocartilage or a combination of fibrocartilage and hyaline-like cartilage (ClinicalTrials.gov Identifier: NCT02855073) [59]. A similar RCT was carried out by Kim et al. (2020), but in this case, regenerative medicine was concomitantly used with high tibial osteotomy (HTO). The aim was to examine whether implanting autologous AD-MSCs alone or in combination with allogeneic cartilage from fresh cadavers can be superior for the treatment of patients with knee OA, and evaluations of clinical improvement and cartilage regeneration showed better results with the combination of autologous AD-MSCs and allogeneic cartilage [60].

Other RCTs attempted to investigate tissue engineering approaches without surgical interventions. Garza et al. (2020) administered an intra-articular injection of autologous stromal vascular fraction (SVF), isolated from adipose tissues by small liposuction harvest, into 39 patients having symptomatic knee OA. In this RCT, two doses of SVF were compared with placebo injections, and results showed a dose-dependent improvement in reducing symptoms and pain, but unfortunately, no changes in cartilage thickness after a 12-month follow-up period were observed (ClinicalTrials.gov Identifier: NCT02726945) [61]. In a similar RCT, the safety and efficacy of intra-articular injections of allogeneic haMPCs were examined. In their work, Lu et al. (2020) evaluated two injections of three different doses of allogeneic haMPCs in 19 patients with symptomatic, bilateral knee OA, and despite the clinical improvement, only the low-dose group showed slight articular cartilage regeneration [62].

Lastly, one case series by Yoon et al. (2020) involving seven patients with ICRS Grade 3 or 4 chondral lesions in the knees examined the utility of costal chondrocyte-derived pellet-type (CCP) as a cell source for ACI. The chondrocytes acquired from costal cartilage were expanded to obtain a 3D pellet for implantation, and a 5-year follow-up showed good clinical improvement, as well as articular cartilage regeneration (ClinicalTrials.gov Identifier: NCT03517046) [63].

In summary, and as shown in Table 1 and Table 2 below, there exists an immense number of cell-based tissue engineering strategies evaluated until date. This information overload, as well as the mixed results and the absence of follow-up studies (especially preclinical in vivo and clinical studies), explains the lack of consensus regarding the best cell-based cartilage tissue engineering strategy. Moreover, the success of stem cells is still questionable, since their superiority over chondrocytes is not proven yet, and their utility without physical and chemical cues did not yield desirable outcomes. All of the uncertainties mentioned before are not aiding in directing the efforts of researchers in the right direction. However, despite the promising results of scaffold-free strategies (implants and injections), we envision the crucial role of scaffolds in supporting cells and delivering cues that are critical for cartilage regeneration.

### 4.2. Cell-Free Tissue Engineering Strategies

Despite their pivotal role in stimulating articular cartilage regeneration, cells require additional considerations when used in tissue engineering, including the surgical procedure for harvesting autologous cells and the time required for cell expansion in vitro. Moreover, it should be mentioned that chondrocytes hold a poor capacity of expansion in vitro, which often limits their use in combination with scaffolds [64]. As a result, researchers brought forward cell-free tissue engineering strategies with the ability to recruit native (endogenous) progenitor or mesenchymal cells as alternatives for cell-based ones. In this section, cell-free tissue engineering is divided into two categories: scaffold-based strategies and injectables.

#### 4.2.1. Scaffold-Based Strategies

For cell-free, scaffold-based articular cartilage regeneration, collagen type I (Col-1)-based scaffolds are mostly applied. The latest evaluation of cell-free Col-1-based scaffolds was conducted by Szychlinska et al. (2020) in vivo using Wistar outbred rat models with knee cartilage lesions at the femoropatellar groove, and this scaffold exhibited biocompatibility and efficient recruitment of host cells for articular cartilage regeneration [65]. Prior research also examined Col-1-based scaffolds, but under different experimental conditions. For instance, Gavenis et al. (2012) performed an in vivo study on Goettinger minipigs with three full-thickness chondral defects of different sizes in the knees to determine the most suitable scaffold size for cartilage repair. To achieve their objective, cell-free Col-1-based scaffold plugs of 8, 10, and 12 mm of diameter were pushed into the defects and fixed with fibrin glue, and the results showed that all three plugs facilitated cellular in-growth and the synthesis of hyaline-like cartilage [66]. A comparable work was formerly conducted by Schneider et al. (2011), but in their in vivo study, they implanted in one group of Goettinger minipig models with full-thickness chondral defects the Col-1-based scaffolds seeded with autologous chondrocytes, and the outcomes revealed that cell-free scaffolds were as effective as cell-based ones in filling chondral defects with hyaline-like cartilage after a 1-year follow-up [67].

Alternatively, different types of scaffolds or biomaterials were also examined by other researchers for cell-free applications. A recent work by Zhiang et al. (2020) involved the fabrication of SF microparticles coated with N-(2-aminoethyl)-4-(4-(hydroxymethyl)-2-methoxy-5-nitrosophenoxy) butanamide (NB) to be applied as a ready-to-use tissue-adhesive for cartilage regeneration in early and middle stages of OA. Studies evaluating this novel joint surface paint (JS-Paint) were conducted in vitro, ex vivo using partial-thickness defect of pig cartilage, and, finally, in vivo using New Zealand white rabbit models with partial-thickness cartilage defects; the results successfully supported JS-paint’s adhesive and chondrogenic properties in all the tested conditions [68]. Another recent work involved the application of cell sheet technology for fabricating biological scaffold-like 3D constructs rather than assembling cell sheets. In their investigation, Wang et al. (2020) used cell sheet technology to fabricate a biological ECM scaffold derived from allogeneic BM-MSCs for osteochondral reconstruction. For this purpose, cell sheets were formed, sodium dodecyl sulfate (SDS) was used for decellularization, and, finally, decellularized ECM scaffolds were obtained. Evaluations of the scaffolds in vitro and in vivo using New Zealand white rabbit models with osteochondral defects showed that the 0.5% SDS treatment is optimal for the simultaneous regeneration of well-vascularized subchondral bone and avascular articular cartilage [69]. Synthetic scaffolds were also implemented for cell-free cartilage tissue engineering. As an example, Dai et al. (2018) built up a novel PLGA scaffold possessing radially oriented microtubular pores. In vitro studies showed that this scaffold allows the migration and distribution of BM-MSCs better than random PLGA scaffolds, and in vivo ones using rabbit models with osteochondral defects confirmed these findings and also showed the simultaneous regeneration of cartilage and subchondral bone [70]. On the other front, Zhang et al. (2015) proposed nanotechnology for the synthesis of cell-free scaffolds for cartilage tissue engineering. In their work, they utilized a hybrid hydrogel composed of Col-2, HA, and PEG and then incorporated magnetic nanoparticles. Evaluations showed that these scaffolds exhibit structural integrity, respond to external magnetic stimuli and travel to target sites, and finally, are cytocompatible according to in vitro studies with Sprague–Dawley rat BM-MSCs [71]. Prior work by Lebourg et al. (2013) evaluated HA/PCL scaffolds, and in vivo assessments using New Zealand white rabbit models with chondral defects conveyed the role of HA in enhancing articular cartilage regeneration avoiding the pre-seeding with cells [72]. Recently, an innovative biomaterial that replicates the mechanical properties of articular cartilage (boundary lubrication and biphasic lubrication) was developed by Milner et al. (2018). In their work, a triple network hydrogel consisting of a double network biphasic hydrogel (water phase and polymer phase) and a biomimetic boundary lubricant (consisting of poly(2-methacryloyloxyethyl phosphorylcholine) (PMPC)) was synthesized and tested in vitro, and the outcomes revealed that this hydrogel possesses superior frictional properties and can prevent opposing chondral damage in partial joint repair [73].

Cell-free scaffolds were further improved by other researchers, who employed them in combination with the delivery of bioactive molecules. In one study by Lolli et al. (2019), a fibrin/HA scaffold was loaded with a microRNA inhibitor targeting miR-221 (antimiR-221) with or without a lipofectamine carrier. This system was assessed both in vitro and in vivo using calves with osteochondral defects, and the obtained results showed that miR-221 silencing in infiltrating cells via this system, especially with the lipofectamine carrier, enhances endogenous repair of osteochondral defects [74]. Another bioactive molecule aimed at cell-free articular cartilage regeneration was proposed by Yu et al. (2015), who tested recombinant human stromal-cell-derived factor 1α (rhSDF-1α) infused into fibrin/HA scaffold as a chondrogenic progenitor cell (CPC) chemoattractant. The outcomes of in vitro and ex vivo studies using full-thickness bovine chondral defects verified both the construct’s cytocompatibility and its promotion of neocartilage generation with proper mechanical properties comparable with the naïve tissue (hyaline cartilage) [75].

Nevertheless, the delivery of growth factors via cell-free scaffolds was more extensively studied over the past few years. An example of such research was conducted by Crecente-Campo et al. (2017), who combined two growth factors, TGF-β3 and BMP-7, and loaded them as nanocomplexes to PLGA scaffolds. When AD-MSCs were cultured on these scaffolds in vitro, the biochemical stimulation provided by the growth factors resulted in an efficient stimulation of articular cartilage regeneration [76]. A similar work was done by other researchers, but as an augmentation to microfracture. Kim et al. (2015) fabricated various scaffolds utilizing modified HA fibers, PCL fibers, and TGF-β3 and combined them as composites to treat microfractures. In addition to confirming the role of TGF-β3 in stimulating articular cartilage regeneration, in vivo studies using Yucatan minipig models with chondral defects also showed that the mechanical properties of HA-based scaffolds were better than those of PLC-containing scaffolds when combined with microfracture [77].

The delivery of growth factors via cell-free scaffolds was also tested in osteochondral regeneration. Lee et al. (2014) evaluated composite scaffolds of hydroxyapatite and PCL loaded with TGF-β3 or mixed with TGF-β3-free collagen solution. To assess these systems in vivo, New Zealand white rabbit models with defective proximal humeral joints were used, and the concomitant regeneration of cartilage and bone, as well as the superiority of TGF-β3-infused scaffolds were evident from the results [78]. Other researchers incorporated more than one growth factor in scaffolds. For instance, Lu et al. (2014) embedded an oligo(poly(ethylene glycol) fumarate) (OPF)-based scaffold in a spatially controlled manner with gelatin microparticles loaded with insulin-like growth factor-1 (IGF-1) and/or BMP-2 and tested the three complexes in vivo using New Zealand white rabbit models with osteochondral defects. The aftermath of the studies showed that BMP-2 specifically enhanced subchondral bone formation, and the synergistic effects of both growth factors may improve subchondral bone but not cartilage regeneration [79]. Similarly, two other growth factors were tested by Re’em et al. (2012), who tested a combination of TGF-β1 and BMP-4 embedded in a bilayer alginate scaffold. Assessments performed in vitro and in vivo using New Zealand white rabbit models with osteochondral defects exhibited complementary results regarding the differentiation of cells into the appropriate cell lineage depending on the biological cues [80].

The controlled biomaterial-mediated delivery system was also tested for the delivery of MSC-derived exosomes (MSC-Exos). The latter surfaced following the novel discovery of the role of MSC-Exos in tissue regeneration and their potential application in articular cartilage regeneration. Accordingly, Jiang et al. (2021) explored hWJ-MSC-derived exosomes (hWJ-MSC-Exos) in combination with porcine-derived acellular cartilage ECM scaffold for the regeneration of knee osteochondral defects. Following the satisfactory cytocompatibility studies in vitro, further in vivo studies using animal models with knee osteochondral defects were performed as follows: Sprague–Dawley rat models received hWJMSC-Exos injection to examine the regulatory effects on the articular cavity microenvironment, whereas New Zealand white rabbit models received hWJMSC-Exos embedded in the acellular cartilage ECM scaffold implant to study the reparative effects. These evaluations revealed the anti-inflammatory and osteochondral regeneration effects of hWJMSC-Exos [81].

#### 4.2.2. Injectables

Since scaffolds are broadly applied for cell-free articular cartilage regeneration, efforts were also exerted to move towards minimally invasive or non-invasive injectable materials and composites. To start with, the application of SF as a cell-free injectable was tested by Yuan et al. (2021) in vitro and in vivo using mice for subcutaneous injection and using New Zealand rabbit models with osteochondral defects. A novel one-step ultrasonication crosslinking method was used for this purpose, which exhibited both safety and efficacy for articular cartilage regeneration [82]. A simple acellular injectable was assessed by Schaeffer et al. (2020) for the treatment of knee osteochondral defects in vivo using Sprague–Dawley rat models, where a microporous annealed particle (MAP) gel was injected and photo-annealed afterwards. When compared to saline injections, MAP hydrogels exhibited stable integration into the defects, providing a promising pro-osteochondral regeneration stimulation, too [83]. Other researchers focused their work on nanotechnology for synthesizing and characterizing nanocomposite-injectable hydrogels. For instance, previous work by Buchtová et al. (2013) was based on siloxane derived hydroxypropylmethylcellulose (Si-HPMC) interlinked with mesoporous silica nanofibers, and the outcomes revealed that this nanocomposite hydrogel exhibited tunable mechanical features and in vitro cytocompatibility with human AD-MSCs and chondrosarcoma cells [84].

On the other hand, researchers applied cell-free injectables as delivery systems for bioactive molecules. Tang et al. (2021) used a thermosensitive poly(d,L-lactide)-poly(ethylene glycol)-poly(d,L-lactide) (PLEL) hydrogel (PDLLA-PEG-PDLLA triblock) and heparin/ε-poly-L-lysine (EPL) nanoparticles incorporated with platelet lysate (PL), which is known to be rich in growth factors, to fabricate composite hydrogels for delivering PL. In vitro evaluations exhibited proper mechanical properties and cytocompatibility, while in vivo studies using osteoarthritic and osteochondral Sprague–Dawley rat models showed satisfactory articular cartilage protection in early-stage OA and regenerative properties in late-stage OA [85]. Another study by Wu et al. (2020) aimed at incorporating an injectable HA hydrogel with PLGA microspheres co-encapsulating KGN and stromal cell-derived factor-1 (SDF-1), and after assessing this system in vitro and in vivo using New Zealand white rabbit models with full-thickness articular cartilage defects, it was evident that this system was able to provide synergistic cell homing and chondrogenic stimulation for repairing articular cartilage defects [86].

#### 4.2.3. Clinical Studies

In parallel to the clinical evaluation of cell-based tissue engineering for articular cartilage regeneration, cell-free tissue engineering was also evaluated by a number of clinical studies, but mainly as an augmentation to microfracture. In a recent prospective cohort study, Cole et al. (2021) augmented microfracture by a cell-free matrix that is a commercial cartilage allograft ECM to act as a scaffold supporting cells released following microfracture. A total of 48 patients with symptomatic focal cartilage defects in the knees were included in this study, and the results evidenced desirable clinical outcomes following the augmented microfracture at 2-year follow-up [87]. A similar method was evaluated in an RCT by Wolf et al. (2020), who evaluated microfracture in combination with a photoreactive chondroitin-sulfate/PEG HA hydrogel. They included in their clinical study 18 patients with full-thickness femoral condyle defects in the knee and followed them up for 24 months after surgery. Not surprisingly, the hydrogel exhibited biocompatibility and high efficacy by improving the articular cartilage defect structural remodeling (ClinicalTrials.gov Identifier: NCT01110070) [88]. Microfracture augmentation was also performed by Lee et al. (2020), but in this case, the authors treated osteochondral lesions by applying an adjunct composed of atelocollagen, thrombin, and fibrinogen. For this purpose, 60 patients with osteochondral lesion of the talus (OLT) participated in the study, and although clinical improvement was observed in both groups (control and investigational), the quality of regenerated cartilage was better with microfracture with atelocollagen augmentation (ClinicalTrials.gov Identifier: NCT02519881) [89]. Another RCT was carried out by Kim et al. (2020), who compared microfracture to porcine-derived collagen-augmented chondrogenesis technique (C-ACT) in 100 patients with cartilage defects in the knee, including those with knee OA. After a 24-month follow-up, it was evident that porcine-derived C-ACT resulted in a better filling of the articular cartilage defects (ClinicalTrials.gov Identifier: NCT02539030) [90].

On the other hand, only a few earlier investigations evaluated cell-free tissue engineering grafting procedures alone. Roessler et al. (2015) performed a prospective case series on Col-1-based scaffolds by recruiting 18 patients with symptomatic articular cartilage defects of ICRS grades 3 and 4, displaying defect size < 11 mm in diameter. The cell-free Col-1-based scaffolds were directly implanted into the defects only in a press-fit manner, and after a 4-year follow-up, results showed improvements in both clinical and imaging scores [91]. Then, the same study was continued and included a series of 28 patients and aimed to provide mid-term data on the efficacy of cell-free Col-1-based scaffolds. However, at 5-year follow-up, the results were not promising because of the reported increase in wear of the repair tissue and the clinical failure in 18% of the participants [92]. A similar experimental study was conducted previously by Efe et al. (2012), where cell-free Col-1-based scaffolds were implanted by press-fit in 15 patients with the same articular cartilage defect pattern described in the previous study, and the results exhibited satisfactory outcomes at the clinical and imaging levels at 2-year follow-up [93]. Furthermore, Schüttler et al. (2014) performed a prospective case series on Col-1-based scaffolds by recruiting 15 patients also suffering from the same articular cartilage defect pattern. Likewise, the cell-free Col-1-based scaffolds were directly implanted by press-fit only, and after a 4-year follow-up period, the results showed improvements in both clinical and imaging scores [94]. In addition, a currently undergoing prospective cohort study by Gupta et al. (2021) is studying the safety and efficacy of umbilical-cord-derived Wharton’s Jelly (UC-derived WJ) as a source of pro-regenerative biochemical factors for the treatment of patients with KL Grades 2 and 3 knee OA. (ClinicalTrials.gov Identifier: NCT04719793) [95].

In summary, and as shown in Table 3 and Table 4 below, most of the cell-free strategies were on Col-1-based scaffolds, and only recently are new biomaterials being assessed. However, these strategies are relatively new, and as with cell-based strategies, there exists conflicting data on the efficacy of these approaches, and there is lack of confirmatory preclinical in vivo and clinical studies. This incomplete picture is the reason why cell-free strategies are not considered to have a better performance in comparison with cell-based ones. Moreover, the success of delivery materials and composites without chemical cues is still questionable, since relying only on their ability to recruit native cells did not yield desirable outcomes. Once again, these uncertainties are not aiding in directing the efforts of researchers in the right direction.

## 5. Conclusions

In spite of being a relatively modern research area, articular cartilage tissue engineering has encountered major advancements over the past two decades, but there are still several obstacles that are hindering its clinical translation. It is unfortunate that translational research is still incapable of bridging the gap between basic research (bench) and clinical research (bedside), and the lack of reproducibility and translatability in the “bench-to-bedside” process is clearly evident with tissue engineering, including articular cartilage tissue engineering. Below, the major data gaps are briefly discussed.

First, there is a general deficiency of knowledge relating to the age-related changes in articular cartilage at the cellular and ECM levels, including zonal alterations as a function of age; in addition to that, the mechanisms and signaling pathways associated with chondrocyte death are not known yet, and possibilities include apoptosis due to dysfunction in the endoplasmic reticulum, dysfunction in the mitochondria, or excessive production of reactive oxygen species (ROS) [3]. Another overlooked factor is the absence of clear distinction between changes in articular cartilage due to physiological aging and those that are related to OA progression [3]. This distinction is of utmost importance since OA is a multifactorial joint condition, and therefore, cartilage defects associated with OA may require multiplex tissue engineering strategies compared to those associated with physiological aging. Second, the ICRS [16] and Outerbridge [96] classifications of articular cartilage lesions were designed for knees and rely on arthroscopic observations, and since they are the only currently available tools, they are applied for other joints and also for magnetic resonance imaging (MRI) observations [97]. Moreover, they do not provide correlations between age-related changes at the cellular and ECM levels and the stages of cartilage damage, and as a result, tissue engineering strategies in ex vivo, in vivo, and clinical studies, especially for the articular cartilage lesions preceding the development of OA, seem to be randomly allocated as they utilize models with physical defects regardless of the environment of the surrounding cartilage tissues. Third, there is a limited number of large-scale clinical trials and long-term follow-up data due to several factors including the novelty of the field for orthopedic surgeons and the difficulty in controlling variables, which highlights the need for the replication of in vitro, ex vivo, and in vivo studies in clinical settings. Clinical trials should also consider determining whether the recovery is complete and is a long-term one, rather than focusing on the improvement in clinical symptoms and radiologic assessments [13,98]. In addition to the aforesaid, several issues arise from the studies available nowadays. For cell-based tissue engineering, the significant setback stems from the large number of studies and the broad spectrum of tissue engineering components and technologies applied [99]. For cell-free tissue engineering, there exists three major challenges: first, the low number of studies; second, the focus is mainly limited to type I collagen scaffolds; and third, the absence of data on its underlying mechanisms of action and its advantages over cell-based tissue engineering and other existing interventions [100]. It is, however, clear that tissue engineering strategies that replicate the niche or the microenvironment of articular cartilage are more likely to exhibit efficient cartilage regeneration, yet it is also crucial to take into account other factors that directly affect the scalability and clinical translation of these strategies including synthesis process, handling in clinics and operation rooms, and ethics. Despite the mentioned limitations, articular cartilage tissue engineering holds great promise in curing and preventing cartilage degeneration.

## Figures and Tables

**Figure 1 materials-15-00031-f001:**
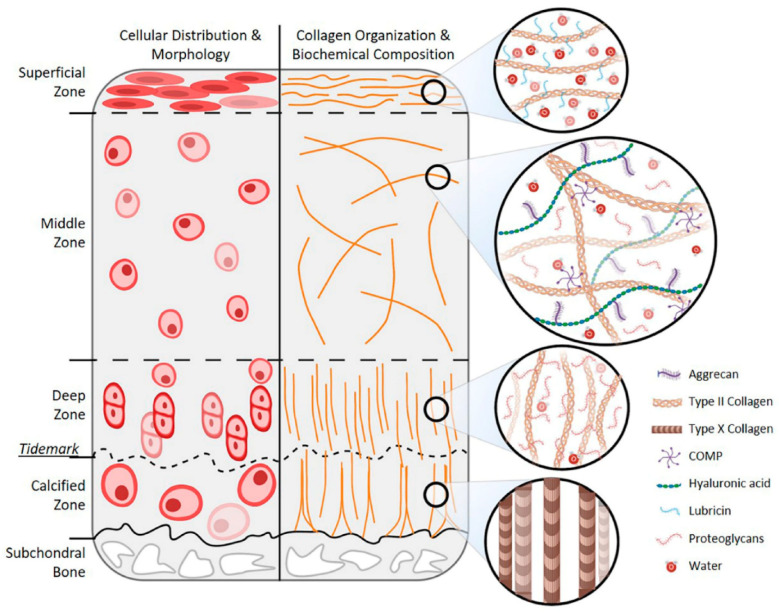
Schematic representation of the articular cartilage structure and biochemical composition [15].

**Figure 2 materials-15-00031-f002:**
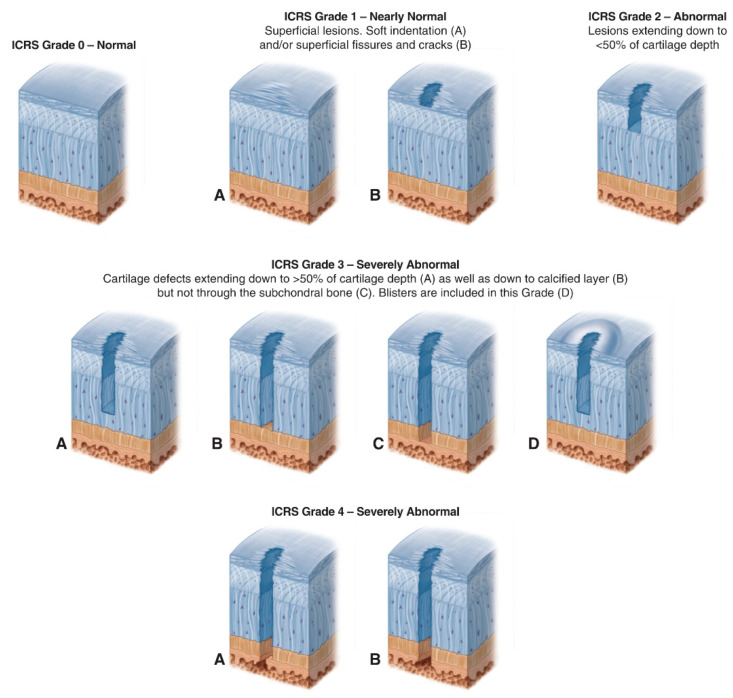
The International Cartilage Repair Society (ICRS) Cartilage Lesion Classification Method. Image reprinted with permission from the ICRS Cartilage Injury Evaluation Package (http://www.cartilage.org/, accessed on the 1 November 2021).

**Table 1 materials-15-00031-t001:** List of studies on cell-based tissue engineering strategies for cartilage defects.

		Cell-Based Tissue Engineering Strategies for Cartilage Defects	Ref.
**01**	**Scaffold-Based Strategies**	MeHA/RGD/HAV + BM-MSCs + KGN in PLGA microspheres	[25]
		PDLLA/GO nanosheets + BM-MSCs + TGF-β3	[26]
		Human dermal-derived collagen + AD-MSCs + collagen substrates	[29]
		SS/SF scaffold + hWJ-MSCs + LAA or PRP	[31]
		SF scaffold + chondrocytes + TAN	[32]
		PCL/RAD16-I + BM-MSCs	[33]
		Fibrin/IEIK13 ± chondrocytes	[34]
		Chitosan/BGP + BM-MSCs/chondrocytes	[35]
		GelMA with bilayered architecture + chondrocytes + TGF-ß1 or mechanical stimulation	[38]
		Gradient PEG-norbornene/PEG-dithiol/CS-MA + chondrocytes or MSCs	[39]
		Bilayered PCL scaffold/porogens + chondrocytes	[40]
**02**	**Scaffold-Free Strategies**	Pre-differentiated BM-MSCs cell sheets	[45]
**03**	**Injectables**	Allogeneic PD cell sheet fragments	[47]
		Autologous AD-MSCs cell sheets	[48]
**04**	**Clinical Studies**	IPFP cell concentrates	[54]
		ACI + CCP	[59]

**Table 2 materials-15-00031-t002:** List of studies on cell-based tissue engineering strategies for cartilage defects associated with osteochondral defects and osteoarthritis (OA).

		Cell-Based Tissue Engineering Strategies for Cartilage Defects Associated with Osteochondral Defects and Osteoarthritis (OA)	Ref.
**01**	**Scaffold-Based Strategies**	Human articular cartilage ECM + AD-MSCs/chondrocytes	[27]
		HA/collagen/fibrinogen + SMSCs + rhTG-4	[28]
		Human placenta + BM-MSCs ± PRP	[30]
		Alginate spheres + chondrocytes/chondrons	[36]
**02**	**Scaffold-Free Strategies**	Human freeze-dried cancellous bone + human chondrocyte sheets	[43]
		Human chondrocytes ± human synoviocytes	[44]
		hAMSCs cell sheets + cartilage particles	[46]
**03**	**Injectables**	AD-MSCs ± HA	[49]
		Open-porous PLGA microspheres + BM-MSCs	[50]
		MSCs + chondrocytes	[51]
		PEG/ApoPep-1/TCO + chondrocytes	[52]
		E7-Exo + SF-MSCs + KGN	[53]
**04**	**Clinical Studies**	Microfracture ± HA ± haMPCs	[55]
		AD-MSCs ± allogeneic cartilage from fresh cadavers	[56]
		Intra-articular injection of autologous SVF	[57]
		Intra-articular injections of allogeneic haMPCs	[58]

**Table 3 materials-15-00031-t003:** List of studies on cell-free tissue engineering strategies for cartilage defects.

		Cell-Free Tissue Engineering Strategies for Cartilage Defects	Ref.
**01**	**Scaffold-Based Strategies**	Col-1-based scaffolds	[61,62,63]
		Col-2/HA/PEG/magnetic nanoparticles	[67]
		HA/PCL scaffolds	[68]
		PMPC/DN biphasic gel	[69]
		Fibrin/HA + rhSDF-1α	[71]
		PLGA + TGF-β3 + BMP-7	[72]
		Modified HA fibers/PCL fibers + TGF-β3 + *microfracture*	[73]
**02**	**Injectables**	Si-HPMC/mesoporous silica nanofibers	[80]
		HA hydrogel + PLGA microspheres co-encapsulating KGN and SDF-1	[82]
**03**	**Clinical Studies**	Commercial cartilage allograft ECM + *microfracture*	[83]
		Photoreactive chondroitin-sulfate/PEG HA hydrogel + *microfracture*	[84]

**Table 4 materials-15-00031-t004:** List of studies on cell-free tissue engineering strategies for cartilage defects associated with osteochondral defects and osteoarthritis (OA).

		Cell-Based Tissue Engineering Strategies for Cartilage Defects Associated with Osteochondral Defects and Osteoarthritis (OA)	Ref.
**01**	**Scaffold-Based Strategies**	SF microparticles coated with NB	[64]
		ECM scaffold derived from allogeneic BM-MSCs	[65]
		PLGA scaffold with radially oriented microtubular pores	[66]
		Fibrin/HA + antimiR-221 ± lipofectamine	[70]
		HA/PCL + TGF-β3 or TGF-β3-free collagen solution	[74]
		OPF-based scaffold /gelatin microparticles + IGF-1 ± BMP-2	[75]
		Bilayer alginate scaffold + TGF-β1 + BMP-4	[76]
		Porcine-derived acellular cartilage ECM scaffold + hWJMSC-Exos	[77]
**02**	**Injectables**	SF injectable	[78]
		Photo-annealed MAP gel	[79]
		PDLLA-PEG-PDLLA + heparin/EPL nanoparticles + PL	[81]
**03**	**Clinical Studies**	Atelocollagen/thrombin/fibrinogen + *microfracture*	[85]
		Porcine-derived C-ACT	[86]
		Col-1-based scaffolds	[87,88,89,90]
		UC-derived WJ	[91]

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
