# Peer review of "Current Advances in the Regeneration of Degenerated Articular Cartilage: A Literature Review on Tissue Engineering and Its Recent Clinical Translation"

_materials, 2021, doi:10.3390/ma15010031_

Round 1

Reviewer 1 Report

The manuscript is a review paper, and it is difficult to assess the validity of statements, without a systematic component to it and a thorough assessment of the available evidence.

Nevertheless, it manages to capture the majority of the papers and ideas in the field, and is therefore interesting as a review. You have to add more on the nano-complex/particles, which are also quite heavily advertised (in research papers) as a possible step forward.

You should also mention Primorac et al group, with autologous injection of the knee, who demonstrated some interesting results. I think it might be good to expand on the failure of stem cells and the possible ways forward, at least as an element of wider discussion.

I liked the link between the lab and clinical studies, and you could also add a sentence or two about the transfer from bench to bed, which is critical item in development of any of these ideas.

Strengthen ideas of the gaps in our knowledge, you could almost make a bidding of the best ideas and ways to move on, for future studies to take on.

Overall, an interesting manuscript, which will become even better after these additions are made.

Author Response

Dear Reviewer 1,

Thank you for your valuable comments. Please find in the document a point-by-point reply. We would like to thank you again for the suggestions that enriched the review.

Reviewer 2 Report

The aim of this review was to highlight current advances in the regeneration of degenerated articular cartilage, including both cell-based and cell-free strategies including in vitro, ex vivo and in vivo models as well as clinical studies and clinical translation. This review also aimed to provide a critical update of the current status of articular cartilage tissue engineering and its clinical translation, relating to cartilage degeneration and OA. The Introduction is well written but the other sections and overall conclusion need major work.

MAJOR COMMENTS

  1. The Materials and Methods are incorrectly described.
  2. The authors state that they always included "Humans" in their literature search. However, they cite research involving the use of non-human derived cells (e.g., bovine, equine such as in citations #37, 38, 40 etc). Please refine/explain.
  3. Also, the authors state that "All English-language experimental, observational, and interventional studies published up to 31 October 2021 were included." This implies that ALL studies from all years are included in the review. However, with the exception of clinical studies, the authors appear to only cite recent papers. Please refine/explain.
  4. The methods applied to clinical publications and studies were not defined and should be defined.
  5. The authors describe a lot of studies but it’s only descriptive. There is no real conclusion from each section
  6. There are not any figures to help summarize the data and to obtain a take-home message, with citations to back it up. Summary figure(s) showing how far the various techniques described have reached regeneration of articular cartilage to help figure out which is the best technique so far (in vitro/ex vivo/in vivo vs. clinically) would be helpful. And/or a figure showing the overall conclusion of which method is better, cell-free or cell-containing scaffolds would be helpful for the readers and for the authors to make a conclusion, which is lacking.
  7. Their overall Conclusion takes conclusions from other reviews. Hence, their conclusions are based on what other papers/reviews state. They need to make their own conclusions from the studies that they addressed in relation to the aim of the review.
  8. The overall Conclusion of the review is not well supported, since they never mention or perhaps don't cite papers that investigate e.g., age-related changes in articular cartilage. The authors nicely discuss this in the Introduction and imply, by including this in the Introduction, that this would be a central part of this review. In the Discussion, they state "There is a general deficiency of knowledge relating to the age-related changes in articular cartilage at the cellular and ECM levels, including zonal alterations". Which studies measured age-related changes? Did the authors include in their methods search for papers that investigated senescent cells, RAGE, AGE, SASPs etc?
  9. Overall conclusion, "...in addition to that, not all the mechanisms and the signaling pathways associated with these changes are clearly tracked down". The authors fail to include any mechanisms and. particularly, the signaling pathway data from the papers cited. Maybe this should go into the conclusion from each section if it truly was not investigated at all in any of these studies?
  10. Overall conclusion, "Another overlooked factor is to properly define an evident distinction between changes in articular cartilage due to physiological aging and those that are related to the OA progression". Which studies investigated regeneration in aged mice vs. OA? Include the citations so that it is clear to the readers which papers investigated aging vs. OA progression.
  11. Overall conclusion, What is the overall conclusion of Cell-Based vs. Cell-Free Tissue Engineering Strategies? Could the authors state which is better? And why? Which regenerates tissue better?
  12. Overall conclusion, "Third, this absence of a detailed classification of articular cartilage lesions is reflected by the relatively random allocation of tissue engineering strategies in ex vivo, in vivo, and clinical studies, especially for the articular cartilage lesions preceding the development of OA. " This sentence is not clear. What does the absence of a detailed classification of articular cartilage lesions have to do with the relatively random allocation of tissue engineering strategies?

MINOR COMMENTS

  1. The authors refer to in vivo models in which the strategies, etc were tested. To understand regenerative effects, it is important to understand the in vivo model in which it was applied. This should be included in the review. The osteochondral models are sometimes included but often the OA or PTOA model (e.g., citations #51, #52, etc.) were not described. Please include/describe.
  2. page 5, line 176: describe what kartogenin is; what does it regulate in MSCs?
  3. Citation #28, define "pro-regenerative pathways". Was this both in vitro and in vivo?
  4. Citation #29, were regenerative effects observed? Define "pro-healing"
  5. line 228, define what tanshinone is.
  6. The authors refer to self-assembling peptides but do not define these peptides. What do they consist of? g., Citation #33, #34
  7. Line 292. THe authors state that "the limitation is due to the evidence that 291 none of the fabricated hydrogels were evaluated via in vivo studies or in clinical trials.". These studies were only published 2018-2021. Therefore, it is not surprising that clinical studies have not yet been performed. Perhaps consider deleting "or in clinical trials."
  8. Citation #45, how were the cells pre-differentiated?

Author Response

Dear Reviewer 2,

Thank you for your valuable comments. Please find in the document a point-by-point reply. We would like to thank you again for the suggestions that enriched the review.

Reviewer 3 Report

The authors of this article have reviewed the literature on regenerative medicine/tissue engineering strategies for treatment of osteoarthritis, with a specific emphasis on the materials side. It is a well-written review and covers a broad range of treatments.

The authors should address the following,

General point

It is advised that the authors have a separate section on cell types and cell-based treatments (PRP, IPFP concentrates) should be included to allow readers to understand a more of the basic biology described in the main sections.

Main points

  1. Introductory paragraph refers to cellular senescence and a general background. This is an important aspect but authors should either remove or move this section to another part of the review. Begin directly on articular cartilage and osteoarthritis.
  2. Under the topic of age-related changes, a focus is on cellular senescence and degeneration. However, the destruction of articular cartilage in OA is multifactorial process that includes changes in biomechances and surrounding tissues. An acknowledgement of this in the context of OA should be included.
  3. Page 7, line 245, “cell-based, scaffold-based articular cartilage”. Should this not be rephrased to cell-scaffold based articular cartilage ?
  4. Page7, in reference to co-culture models, an acknowledgement of the rationale to use chondrocytes-MSCs co-culture needs to be stated. This is specifically, the induction of cartilage hypertrophy in MSC chondrogenesis (Pelttarri et al, 2006, A and R).
  5. For the preclinical in vivo work, it is advised that this should section similar to the in vitro and clinical studies. A table should be created for the different strategies used and their outcomes. It is difficult to know from reading the text, the difference between in vitro and preclinical in vivo studies in the current format.
  6. For the clinical studies, a table describing the different outcomes from each of the studies should be presented with a clear outcomes of their clinical use. This would enable the reader to easily understand the different points described in the main text.

Author Response

Dear Reviewer 3,

Thank you for your valuable comments. Please find in the document a point-by-point reply. We would like to thank you again for the suggestions that enriched the review.

Round 2

Reviewer 2 Report

The manuscript has been improved.

Only minor comments:

1. What exactly KGN is not described. ...hint: In the year 2012, Johnson and his colleagues published the discovery of KGN after screening 22,000 drug-like molecules similar to the natural ligands involved in cell signaling and differentiation. (Johnson 2012 Science). And please add citations for newly added text that "KGN is a chondrogenic and chondroprotective agent".

2. Tanshinone is nicely described (it is an active ingredient extracted from Danshen root (Salvia miltiorrhiza 250 Bunge). But the citations are missing from the remaining part of the sentence are missing "...with anti-inflammatory, antioxidative, and anti-apoptotic properties"...please add appropriate citations to support your statement

Author Response

Dear Reviewer 2,

Thank you very much for pointing these uncertainties out.

  1. Kartogenin: Johnson et al. were cited and their definiton of kartogenin was used. The aim of using kartogenin by Teng et al. was also mentioned to stress on its role.
  2. Tanshinone IIA: Three references discussing the properties of tanshinone IIA were added.

Best Regards.

Reviewer 3 Report

The authors have answered my questions appropriately.

Author Response

Dear Reviewer 3,

The authors would like to thank you again for your comments and suggestions. 

Best Regards.